# Metabolomic Approach to Screening Homozygotes in Chinese Patients with Severe Familial Hypercholesterolemia

**DOI:** 10.3390/jcm12020483

**Published:** 2023-01-06

**Authors:** Zhiyong Du, Yunhui Du, Linyi Li, Haili Sun, Chaowei Hu, Long Jiang, Luya Wang, Yanwen Qin

**Affiliations:** 1The Key Laboratory of Remodeling-Related Cardiovascular Diseases, Ministry of Education, National Clinical Research Center for Cardiovascular Diseases, Beijing Anzhen Hospital, Capital Medical University, Beijing 100029, China; 2Beijing Institute of Heart Lung and Blood Vessel Disease, Beijing 100029, China; 3Department of Cardiology, The Second Affiliated Hospital of Nanchang University, Nanchang 330006, China

**Keywords:** homozygous familial hypercholesterolemia, severe heterozygous familial hypercholesterolemia, metabolomics, diagnostic markers

## Abstract

Homozygous familial hypercholesterolemia (HoFH) is a rare inborn-errors-of-metabolism disorder characterized by devastatingly elevated low-density lipoprotein cholesterol (LDL-C) and premature cardiovascular disease. The gold standard for screening and diagnosing HoFH is genetic testing. In China, it is expensive and is always recommended for the most likely HoFH subjects with aggressive LDL-C phenotype. However, the LDL-C levels of HoFH patients and a substantial proportion of heterozygous FH (HeFH) patients overlapped considerably. Here, we performed a cost-effective metabolomic profiling on genetically diagnosed HoFH (*n* = 69) and HeFH patients (*n* = 101) with overlapping LDL-C levels, aiming to discovery a unique metabolic pattern for screening homozygotes in patients with severe FH. We demonstrated a differential serum metabolome profile in HoFH patients compared to HeFH patients. Twenty-one metabolomic alterations showed independent capability in differentiating HoFH from severe HeFH. The combined model based on seven identified metabolites yielded a corrected diagnosis in 91.3% of HoFH cases with an area under the curve value of 0.939. Collectively, this study demonstrated that metabolomic profiling serves as a useful and economical approach to preselecting homozygotes in FH patients with severe hypercholesterolemia and may help clinicians to conduct selective genetic confirmation testing and familial cascade screening.

## 1. Introduction

Familial hypercholesterolemia (FH) is a typical inborn-error-of-metabolism (IEM) disorder caused by pathogenic variants in several genes involved in the metabolism of low-density lipoprotein cholesterol (LDL-C) [1,2]. The pathogenic variants in the gene encoding low-density lipoprotein receptor (LDLR) are the most common. Less frequently, mutations in genes encoding apolipoprotein B (APOB), proprotein convertase subtilisin/kexin 9 (PCSK9), and LDLR adaptor protein 1 (LDLRAP1), are also associated with FH phenotypes [3,4].

The heterozygous genotype of FH (HeFH) is usually caused by a single pathogenic variant, globally occurring in one of every 300–500 individuals [5,6]. Homozygous FH (HoFH) is a rare condition caused by biallelic pathogenic variants, affects one in 160,000–400,000 people worldwide [6,7]. China, as the most populous country in the world, is facing a heavy health and economic burden of FH. In mainland China, the prevalence of HeFH has been estimated to be one in 200–500 individuals [8]. Consequently, the prevalence of HoFH has been predicted to be 1:600,000 [8,9].

HoFH is characterized by extremely high LDL-C levels since birth. If left undiagnosed and untreated, HoFH patients can develop markedly premature atherosclerotic cardiovascular disease (ASCVD) and die at an early age [10,11]. An early and accurate diagnosis of HoFH underlines the importance of achieving optimal outcomes in patients and provides a significant means of conducting further genetic familial-cascade screening for identifying affected familial relatives [12]. Diagnosis of HoFH can be made on the basis of clinical or genetic criteria. While genetic testing can provide a definitive diagnosis of HoFH [13]. Patients with HoFH usually receive statins in combination with additional cholesterol-lowering agents, such as ezetimibe or PCSK9 inhibitors. Lipoprotein apheresis or liver transplantation may be used as alternative options for patients who have poor response to lipid lowering drugs [7,12].

Although genetic testing can accurately identify HoFH, it is expensive and is not available in most clinical settings in developing countries. In China, genetic testing is always performed in index HoFH patients and their familial members [8,14]. The diagnosis of an index HoFH is largely based on the clinical phenotype upon the extremely elevated LDL-C concentrations and family history. This is because most HoFH patients typically present higher LDL-C levels and premature cardiovascular events than HeFH patients and the common hypercholesterolemia population. Notably, a substantial proportion of HeFH cases with severe LDL-C levels might overlap with LDL-C values observed in HoFH individuals [15,16]. Therefore, the clinical criteria based on LDL-C levels might not truly discriminate HoFH from the aggressive form of HeFH. Additional and comprehensive evaluation by other HoFH-related clinical indicators (such as extensive xanthomas, aortic stenosis, corneal arcus) are always warranted.

Untargeted metabolomics with broad metabolome coverage and rapid biomarker detection capacity is increasingly utilized in the clinical screening and diagnosis of IME [17]. Since untargeted profiling serves as an innovative approach to provide a comprehensive metabolic fingerprint, it offers an unprecedented opportunity for better understanding the effects of genetic variations in the biochemical phenotype of individuals with IME disorders [18,19]. Herein, using liquid chromatography/mass spectrometry (LC/MS)-based untargeted metabolomics, we sought to systematically explore the serum metabolome landscapes of patients with genetically diagnosed HoFH and HeFH who had overlapping LDL-C values and test whether implementation of metabolomic profiling could facilitate the detection of homozygotes in FH patients with aggressive LDL-C phenotype.

## 2. Materials and Methods

### 2.1. Study Population

All patients were enrolled from the Familial Hypercholesterolemia Families Cohort (FHFC) affiliated to Beijing Anzhen Hospital of the Capital University of Medical Sciences between January 2018 and July 2022 [9,14]. This study cohort has been registered with www.chictr.org.cn/index.aspx (number: ChiCTR1900022156). The study protocol was conducted in accordance with the Declaration of Helsinki and approved by the Ethics Committees of Beijing Anzhen Hospital of the Capital University of Medical Sciences. All participants provided written informed consent, which was the review procedure of the ethics committee. FH diagnosis was established by means of the Dutch Lipid Clinic Network (DLCN) diagnostic criteria [20]: DLCN score ≥ 6, an untreated LDL-C ≥ 4.7 mmol/L, either corneal arcus or xanthomas, premature ASCVD history in first-degree relatives, and genetically confirmed FH. The diagnostic criteria for HoFH were as follows: untreated LDL-C ≥ 13 mmol/L or treated LDL-C ≥ 8 mmol/L, either cutaneous or tendon xanthomas, and two mutations in the LDLR, APOB, PCSK9, or LDLRAP1 genes [14,20].

Patients were excluded if they received regular lipoprotein apheresis therapy or liver transplant surgery. Finally, a total of 67 HoFH patients and 119 HeFH patients with overlapping LDL-C concentrations were included in the metabolomic study. All study subjects were genetically confirmed to have mutant alleles at the LDLR gene. Genotyping was obtained from patients if it had been performed or tested according to our previously reported methods [14,21].

### 2.2. Sample and Clinical Data Collection

Blood samples were drawn at the time of enrollment after an overnight fast. Sera were separated by centrifugation at 1300× *g* for 20 min, and then stored at −80 °C until analysis. The serum levels of LDL-C, total cholesterol (TC), triglycerides (TG), high-density lipoprotein cholesterol (HDL-C), and lipoprotein (a) [Lp(a)] were determined using an automatic biochemistry analyzer AU 5400 (Beckman, Brea, CA, USA). Demographic characteristics, including age, sex, smoking status, lipid-lowering treatments, and chronic diseases (e.g., hypertension, diabetes mellitus) were recorded for each participant.

### 2.3. Sample Preparation for Metabolomic Analysis

Metabolite extraction was followed with our previously established protocol [14,22]: Briefly, a total of 50 μL aliquots of serum was thawed at 4 °C, a volume of 200 μL ice-cold solution (acetonitrile: water = 1:1, *v*/*v*) containing a variety of isotopic internal standards (0.25 mg/L cholic acid-2, 2, 4, 4-*d*_4_, 0.54 mg/L tauroursodeoxycholic acid-*d*_4_, 0.35 mg/L glycocholic acid-*d*_4_, 0.35 mg/L L-leucine-5, 5, 5-*d*_3_, 0.15 mg/L L-arginine-*d*_7_, 0.12 mg/L stearic acid-18, 18, 18-*d*_3_, 0.21 mg/L LysoPC (19:0)-*d*_5_, 0.16 mg/L PC(18:0/20:4)-*d*_11_, 0.20 mg/L cholesterol-*d*_7_, 0.1 mg/L stearoyl-L-carnitine-*d*_3_) was added. The mixture was vortexed for 2 min and centrifuged at 15,000 rpm for 12 min, at 4 °C. A total of 200 μL supernatant was transferred into a clean dry tube and evaporated to dryness. The dried residue was stored at −80 °C. Quality control (QC) samples were prepared by mixing equal aliquots from each sample and processed following the above methods. The dried residue was reconstituted in 100 μL of water: methanol (1:1, *v*/*v*) solution before metabolomic analysis.

### 2.4. Metabolomic Analysis and Data Processing

Metabolomic profiling was performed on an ACQUITY ultra performance liquid chromatography (UPLC) system coupled with a dual electrospray ionization probe and a micro mass quadrupole-time of flight (QTOF) micro synapt high-definition mass spectrometer (Waters Corporation, Milford, MA, USA). The injection volumes of the samples were all 2 μL. Metabolite separation was achieved with an ACQUITY UPLC HSS T3 column (100 mm × 2.1 mm, 1.8 μm, Waters Corporation, Milford, MA, USA). The mobile phase consisted of a linear gradient system of 0.1% formic acid in acetonitrile (A) and 0.1% formic acid in water (B): 0–1.5 min, 0–5% A; 1.5–5.0 min, 5–45% A; 5.0–7.5 min, 45–55% A; 7.5–10.0 min, 55–70% A; 10.0–12.0 min, 70–100% A; 12.0–13.5 min, 100% A; 13.5–15.0 min, 100–0% A. The column was maintained at 35 °C, and the flow rate was set at 0.4 mL/min. Both positive and negative MS modes were performed. The parameters of MS detection were as follows: Capillary voltage was set at 3.2 kV and 2.5 kV for positive mode and negative mode, respectively. The source temperature was set at 110 °C. The sampling cone voltage and cone gas rate were set at 40 V and 50 L/h, respectively. The desolvation gas temperature and desolvation gas flow were 400 °C and 650 L/h, respectively. The MS scanning range was set at 50–1100 Da with a collision energy range from 10 to 55 eV.

The UPLC–QTOF/MS-acquired raw data were extracted, peak-identified and QC-processed using Progenesis QI (Waters Corporation, Manchester, UK). The normalized semiquantitative datasets were calculated by using the isotopic internal standards. Metabolite identification was performed on the Progenesis QI MetaScope, HMDB databases, METLIN database, and in-house metabolite library using the primary and secondary MS information. To gain a comprehensive view of the clustering trends between the study groups, principal component analysis (PCA) of the normalized data matrix was established by using SIMCA-P software (v14.0, Umetrics, Umea, Sweden). To identify the differentially expressed metabolites, a volcano plot was calculated using the Mann–Whitney *U* test and fold changes utilizing MetaboAnalyst (http://www.metaboanalyst.ca/, accessed on 10 November 2022), a false discovery rate (FDR)–calibrated *p* value < 0.05 and fold change (FC) >2 or <0.5 was considered significant.

### 2.5. Statistical Analysis

Categorical variables are presented as frequencies (n) and percentages (%) and were compared by using Chi-square test. Continuous data are presented as the mean and standard deviation (means ± SD), and the data not normally distributed are expressed by medians and interquartile ranges [IQR]. Student’s *t* test and Mann–Whitney *U* test were used for the comparisons of normally and nonnormally distributed data, respectively. The association of differentiated metabolites with HoFH or HeFH was performed by logistic regression model-based association analyses using the value of β-coefficients. FDR–adjusted *p* < 0.05 was considered significant. The classification model of the altered metabolites was established by random forest-based Monte Carlo cross validation (MCCV) and receiver-operating characteristic (ROC) curves. The prediction performances were evaluated by using posterior classification probability (100 cross-validations). All analyses were performed by using MetaboAnalyst, SPSS Statistics 26 (IBM Corp., New York, NY, USA), and the bioinformatics platform (http://www.bioinformatics.com.cn/login/, accessed on 15 November 2022).

## 3. Results

### 3.1. Subject Characteristics

A total of 170 genetically confirmed FH patients were included, including 69 patients with HoFH (male, 50.72%; age, 23.4 ± 15.2 years) and 101 patients with severe HeFH (male, 55.45%; age, 27.1 ± 12.1 years). The demographic characteristics and clinical lipids of the study individuals are depicted in Table 1. There were no significant differences in age, sex, or the prevalence of hypertension, diabetes mellitus, and ASCVD history between HoFH and HeFH patients (all *p* values > 0.05). At the time of enrollment, most participants had received lipid-lowering therapies (LLT), and there was no difference in the LLT options (statins, or the combination of statins and ezetimibe) between the study groups. Furthermore, no significant differences were observed in the circulating levels of LDL-C and TC between them. Notably, patients with HoFH presented elevated levels of Lp(a) and decreased levels of HDL-C and TG compared to patients with the severe form of HeFH (all *p* value < 0.05).

### 3.2. Sera Metabolome Profiles of HoFH and Severe HeFH

Using the rapid UPLC–QTOF/MS platform, we identified a total of 242 hydrophilic metabolites, 436 lipid species, and 130 unknown compounds with less than 20% relative standard derivations (RSD) of peak intensity in the QC samples (Figure 1A). Then, we examined the metabolite datasets globally with an unsupervised PCA score plot, in which the serum samples of HoFH and severe HeFH patients were distributed based on the first two principal components (Figure 1B). Interestingly, although no significant differences were observed in the concentrations of LDL-C and TC between the two study groups, we found that HoFH and HeFH patients had a totally different metabolomic profile.

Volcano plots highlighted 47 differentiated metabolic alterations (21 up, 26 down) between HoFH and severe HeFH patients, including 24 identified metabolites and 23 unknown compounds (Figure 1C). Compared to HoFH patients, deoxycholic acid (DCA), lithocholic acid (LCA), hyodeoxycholic acid (HDCA), ten triacylglycerol (TAG) species, and two polyunsaturated glycerophosphocholine (GPC) species were significantly increased in the sera of HeFH patients, whereas circulating pipecolic acid, 3-phenylpropionate, 3-indole propionic acid, isocitric acid, stearoyl-carnitine, oleoyl-carnitine, arachidoyl-carnitine, and two lysophosphatidic acid (LysoPA) species were decreased. Spearman’s rank-based correlation network of the altered metabolites and clinical lipids (|correlation coefficient| > 0.6 and *p* < 0.05) is depicted in Figure 1D. As expected, TAG species were positively associated with TG. In addition, we found that LysoPA 18:0, LysoPA 18:1, and oleoyl-carnitine were positively correlated with Lp(a), HDCA and two unknown compounds were significantly associated with HDL-C.

### 3.3. Specific Biosignatures for Differentiating HoFH from Severe HeFH

To evaluate the contribution of the metabolomic alterations to the discrimination of HoFH from severe HeFH, we conducted univariate and multivariable regression analyses. In univariate analysis (Figure 2A), most of the altered metabolic features (41 in 47) statistically contributed to the classification of HoFH and severe HeFH (FDR-adjusted *p* < 0.05). After adjustments for ages, gender, LLT options, comorbid diseases, and all clinical lipid measures, we found that 21 metabolomic features still exhibited significant performances in differentiating HoFH from severe HeFH, including 14 unknown compounds and seven identified metabolites, namely, LCA, TAG (52:2), 3-phenylpropionate, pipecolic acid, 3-indolepropionic acid, isocitric acid, and GPC (38:5).

Compared to LDL-C, the seven identified differential metabolites showed remarkable performance in distinguishing HoFH from severe HeFH. The univariate ROC of each individual metabolite yielded an area under the ROC (AUC) value > 0.81 (Figure 2C). Furthermore, the random forest algorithm-based MCCV model indicated that the combination of seven altered metabolites achieved a maximized performance in discriminating HoFH from severe HeFH (AUC value = 0.939, Figure 2D). Using posterior classification probability-based prediction model, we found that the combination of seven metabolites could correctly classify 91.3% (63 in 69) of patients with HoFH and 90.1% (91 in 101) of patients with HeFH (Figure 2E). These results demonstrated that untargeted metabolomics facilitated the accurate discrimination of HoFH patients from HeFH patients with severe and overlapping levels of LDL-C.

## 4. Discussion

Early diagnosis is of great significance for preventing ASCVD and improving the life expectancy of HoFH patients by initiating early treatment. In developed countries, such as the Netherlands, genetic test-based family cascade screening has been shown to be effective in reducing premature morbidity and mortality in the HoFH population [13,23]. In mainland China, genetic testing, including limited-variant array and comprehensive next-generation genetic sequencing, are expensive and self-paying procedures, which cost approximately USD 750–1400. Therefore, they are always recommended in the most likely HoFH patients with an aggressive phenotype of LDL-C levels [24]. However, emerging clinical evidences indicated that many HeFH patients might also exhibit an extremely elevated level of LDL-C [16,25]. China has a population of approximately 1.4 billion people, accounting for 8% of all FH patients in the world [8,24,26]. Thus, performing genetic testing on suspected HoFH cases identified by LDL-C level-based criteria remains a huge economic challenge in China.

Over the last decade, metabolomics, as a new clinical laboratory tool, has been widely used for the detection of a variety of rare IEM diseases [27,28]. In this study, using rapid and high-throughput LC/MS metabolomic technique, we identified a panel of 21 metabolomic alterations that were independently associated with HoFH condition. We found that HoFH could be correctly preselected in patients with aggressive phenotypes of LDL-C levels by using seven identified metabolite signatures-based diagnostic algorithm, even without genetic tests. Currently, LC/MS-based metabolomic platforms are available in most Grade III Level A hospitals across mainland China. Furthermore, compared with genetic testing, clinical metabolomics is a more economical measure that cost approximately USD 100–120. This work demonstrates that the established metabolomic approach may not only provide promising opportunities to accurately differentiate HoFH from the severe form of HeFH in clinical practice, but also offers a more cost-effective option for patients and clinicians.

It is well recognized that two mutated alleles in HoFH produce a higher elevation of LDL-C than one mutated allele in HeFH [29]. However, many potential impacts such as high-fat diets, gut microbes, drugs, or other biochemical abnormalities have also been reported to substantially influence the levels of LDL-C in HeFH individuals [15]. In this study, several gut microbiota-derived metabolites were found to be significantly altered in the sera of patients with severe HeFH compared to HoFH patients, including decreased 3-indolepropionic acid and three increased secondary bile acids (including DCA, LCA, and HDCA). 3-indolepropionic acid is a product of microbial tryptophan metabolism, and its decrease has been reported to be associated with lipid accumulation, especially in triacylglycerol (TAG) species [30]. Interestingly, our results showed that HeFH patients exhibited increased levels of TAG species and decreased levels of 3-indolepropionic acid compared to HoFH patients. TAG is an important basic component of low-density lipoproteins. Our findings suggested that the gut microbiota-derived 3-indolepropionic acid might play a potential role in the low-density lipoprotein metabolism via regulating the TAG levels.

Secondary bile acids are the major types of bacterial metabolites, which are converted from primary bile acids by gut microbiota [31]. Numerous evidences have demonstrated that bile acids and the gut microbiota in the intestine could regulate the digestion and absorption of cholesterol and triglycerides, but also play a key role in modulating lipid metabolism by activating farnesoid X receptor [32,33]. The altered secondary bile acids in the blood might act as an indirect readout for the abnormal gut microbiome in patients with severe HeFH. Further studies investigating the microbial alterations in the severe form of HeFH and the mechanism of action of secondary bile acids in regulating circulating cholesterol homeostasis are warranted. Another important finding of this study is that HoFH patients exhibited higher levels of Lp(a) and its metabolic products (LysoPA species) than patients with severe form of HeFH. Previous studies have demonstrated the negative functions of Lp(a) and LysoPA in promoting ASCVD progression and regulating inflammation [34,35]. Therefore, patients with HoFH might have an increased ASCVD risk than HeFH patients with an aggressive phenotype of LDL-C levels.

Some limitations warrant discussion. Untargeted metabolomics will still need to resolve some issues (e.g., accurate quantification and unknown compound identification) to become more useful in screening and diagnosing HoFH in clinical practice. Additional experimental data will be needed to explore the fundamental causes for the differentiated metabolite profiles between HoFH and severe HeFH. The participants of this study were Chinese people, which might limit the generalizability of our results to other ethnic populations.

## 5. Conclusions 

Collectively, this study demonstrates that untargeted metabolomic profiling offers a cost-effective approach for detecting HoFH in Chinese FH patients with a severe LDL-C concentration phenotype. Moreover, integrating this with clinical information may help clinicians accurately preselect an index HoFH patient for conducting subsequent genetic confirmation testing and selective cascade screening of affected family members.

## Figures and Tables

**Figure 1 jcm-12-00483-f001:**
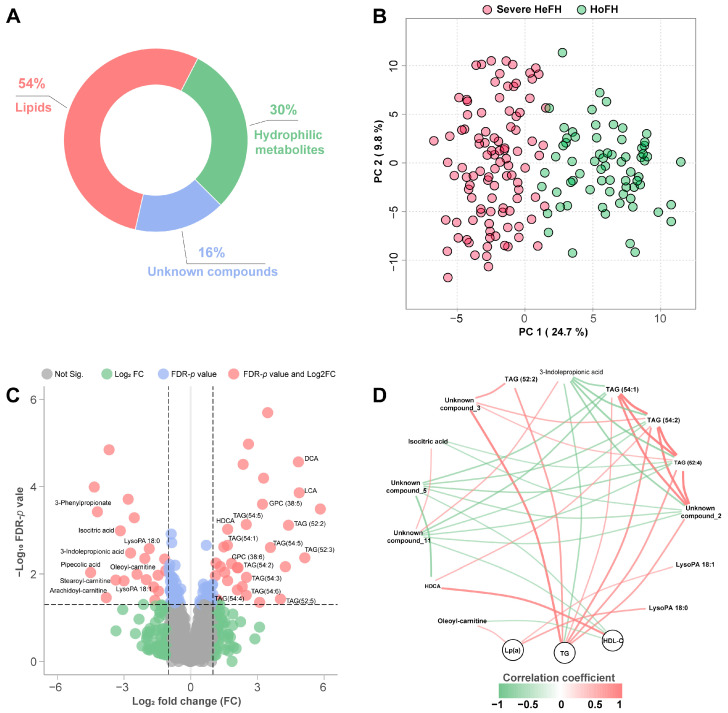
Changes in the serum metabolomic profiles of HoFH and severe HeFH patients. (**A**) Distribution of the metabolomic variables in this study. (**B**) Principal component analysis (PCA) score plot of the serum samples from HoFH and severe HeFH using principal component (PC) 1 and PC2. (**C**) Volcano plots of the metabolomic datasets depicting the differential metabolomic alterations between HoFH and severe HeFH. FDR *p* value < 0.05 and fold change (FC) >2 or <0.5 were considered significant. Not Sig., not significant. (**D**) Spearman’s rank-based correlation network of the altered metabolites and clinical lipids. |correlation coefficient| > 0.6 and *p* < 0.05 were considered significant.

**Figure 2 jcm-12-00483-f002:**
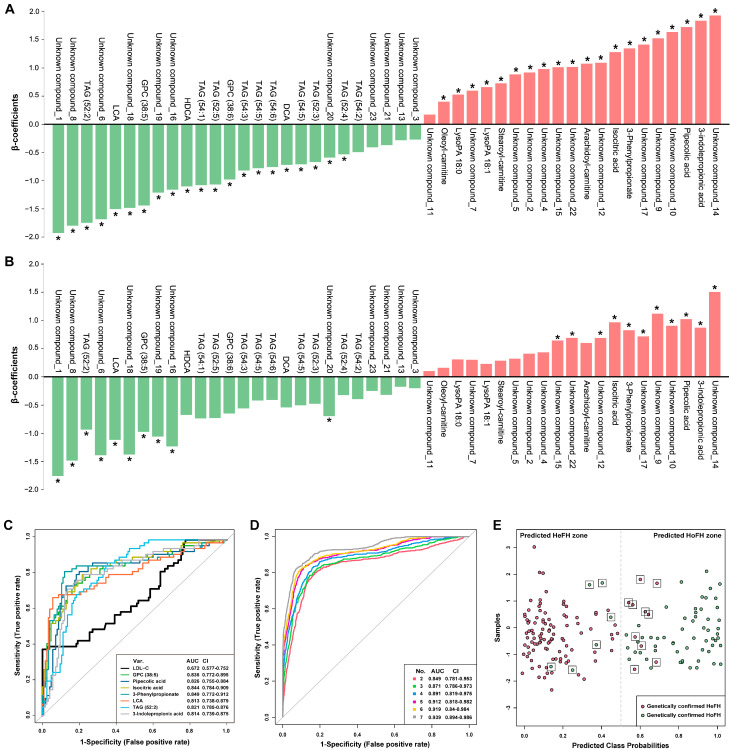
Metabolomic biosignatures for discriminating HoFH from severe HeFH. (**A**,**B**) Univariate (**A**) and multivariable (**B**) regression plots depicting the association of metabolomic alterations with HoFH. A β-coefficient value > 0 indicated a positive association with HoFH and a negative association with HeFH; a β-coefficient value < 0 indicated a negative correlation with HoFH and a positive correlation with HeFH. *: FDR-adjusted *p* < 0.05. Multivariable adjustments included ages, gender, LLT options, comorbid diseases, HDL-C, non-HDL, LDL-C, TC, TG, and Lp(a). (**C**) Receiver operating characteristic (ROC) curve assessing the discriminatory performance of the individual metabolite biosignature. (**D**) Multivariable ROC curve of the multiple metabolite combinations in differentiating HoFH from severe HeFH. (**E**) Posterior classification probability plot showing the classification of HoFH and severe HeFH according to the seven metabolite-based predictive model. Each symbol represents the classification probability that a given sample belongs to genetically confirmed HoFH or HeFH. The samples with an incorrect prediction categorization are marked with black boxes.

**Table 1 jcm-12-00483-t001:** Demographic and clinical characteristics of all subjects.

	HoFH (*n* = 69)	Severe HeFH (*n* = 101)	*p* Values
Ages	23.4 ± 15.2	27.1 ± 12.1	0.076
Male sex, *n* (%)	35, (50.72%)	56, (55.45%)	0.54
Hypertension, *n* (%)	2, (2.90%)	7, (6.93%)	0.25
Diabetes mellitus, *n* (%)	0, (0.0%)	4, (3.96%)	0.094
Current smokers, *n* (%)	1, (1.45%)	8, (7.92%)	0.064
ASCVD history, *n* (%)	5, (7.25%)	2, (1.98%)	0.089
Non-LLT, *n* (%)	6, (8.70%)	11, (10.89%)	0.64
Statins alone, *n* (%)	38, (55.07%)	65, (64.36%)	0.22
Statins and ezetimibe, *n* (%)	25, (36.23%)	27, (26.37%)	0.19
LDL-C, mmol/L	9.39 ± 1.83	8.90 ± 1.95	0.091
TC, mmol/L	11.88 ± 2.27	11.35 ± 2.21	0.14
TG, mmol/L	0.84 [0.60, 1.27]	1.33 [0.87, 1.87]	0.0041
HDL-C, mmol/L	1.04 ± 0.37	1.26 ± 0.29	0.014
Non-HDL, mmol/L	10.86 ± 2.27	10.09 ± 2.24	0.032
LP(a), mg/dL	41.9 [23.3, 56.5]	15.6 [6.6, 19.6]	<0.0001

Continuous data are presented as the mean ± standard deviation or median [interquartile range], and categorical variables are presented as %. Two-tailed Student’s *t* test or Mann–Whitney *U* test were used for continuous data. The Chi-square test was used for categorical data. ASCVD, atherosclerotic cardiovascular disease; LDL-C, low-density lipoprotein cholesterol; TC, total cholesterol; TG, triglycerides; HDL-C, high-density lipoprotein cholesterol; Lp(a), lipoprotein (a); HoFH, homozygous familial hypercholesterolemia; HeFH, heterozygous familial hypercholesterolemia.

## Data Availability

The data are contained within the article.

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
