# Peer review of "Metabolomic Approach to Screening Homozygotes in Chinese Patients with Severe Familial Hypercholesterolemia"

_jcm, 2023, doi:10.3390/jcm12020483_

Round 1
Reviewer 1 Report
This article is well written and it describes a metabolomic approach to HoFH patients detection in China.
I think that the authors must consider and state in the manuscript that the clinical and biochemical presentation of patients with HoFH is often sufficient to distinguish them from patients with severe forms of HeFH. Moreover, patients with severe form of HeFH need genetic testing and family screening as well. Family history for premature cardiovascular events is a cheap and quick method that must be considered in the detection of HoFH and HeFH patients. What is more, the authors do not say what is the difference between the cost of the genetic test for FH (a well known and established nean of diagnosis) and this new experimental approach.
However, I think that from an academic point of view this is an interesting study in the specific setting were it has been performed.
Author Response
Response to Reviewer 1 Comments
Point 1:
This article is well written and it describes a metabolomic approach to HoFH patients detection in China.
I think that the authors must consider and state in the manuscript that the clinical and biochemical presentation of patients with HoFH is often sufficient to distinguish them from patients with severe forms of HeFH. Moreover, patients with severe form of HeFH need genetic testing and family screening as well. Family history for premature cardiovascular events is a cheap and quick method that must be considered in the detection of HoFH and HeFH patients. What is more, the authors do not say what is the difference between the cost of the genetic test for FH (a well known and established mean of diagnosis) and this new experimental approach.
However, I think that from an academic point of view this is an interesting study in the specific setting were it has been performed.
Response 1: Dear reviewer, thank you very much for your critical and thoughtful comments. Your comments are very useful for improving the quality of our article. We fully agreed with your comments. In general, the clinical and biochemical indexes (such as untreated LDL-cholesterol, premature atherosclerosis, extensive xanthomas, corneal arcus, aortic stenosis) are sufficient to distinguish HoFH from severe form of HeFH. As you mentioned, family history for premature cardiovascular events is a cheap and quick method that must be considered in the detection of HoFH and HeFH patients. In the present study, family history for premature ASCVD in first-degree relatives is considered as a very important index for the diagnosis of HoFH and HeFH.
We also do agree with your points “Moreover, patients with severe form of HeFH need genetic testing and family screening as well”. Both of patients with HoFH and patients with severe form of HeFH are at high cardiovascular risk. In the present study, we observed a similar prevalence of ASCVD events between the study HeFH and HoFH patients with overlapped LDL-cholesterol. An early diagnosis underlines the importance of achieving optimal outcomes in these patients and provides a significant means of conducting further family cascade screening for identifying affected familial members.
The gold standard for diagnosing HoFH or HeFH is genetic testing. Theoretically, HoFH patient's family may have more affected relatives compared to HeFH patient's family. However, performing genetic testing on suspected HoFH cases and their family members remains a huge economic challenge in China. Because genetic testing is a cost-expensive and self-paying item, which is not included the national medical insurance of China. The limited-variant array and comprehensive genetic next-generation sequencing cost about $750 and $1400, respectively. In contrast, LC-MS/MS-based metabolomic testing costs approximately 100–120 dollars. As a proof of concept, this study aimed to use the cost-effectiveness metabolomic approach to identify HoFH in patients with serve LDL-cholesterol phenotype and may provide a potentially useful approach to preselecting homozygotes in FH patients with severe hypercholesterolemia and may help clinicians to conduct selective genetic confirmation testing and familial cascade screening.
According to your valuable comments and suggestions, we have rewritten the section of introduction and discussion. All the related changes were marked up using the “Track Changes” in the revise manuscript. Again, we fully thank for your time and attention in our manuscript, and we hope that the revised article can achieve your expectation and will be worthy of publication in Journal of Clinical Medicine.

Reviewer 2 Report
In general, the authors have performed a well written manuscript about an important aspect of clinical cardiology. Interestingly, there is a need for novel biomarkers in order to assess the risk of patients with lipid-disorders. After reading the manuscript I have the following comment to suggest for you:
In the introduction, you could include (in short) the guidelines' recommendations about diagnosis and management of dyslipidemia in these patients.
Author Response
Response to Reviewer 2 Comments
Point 1:
In general, the authors have performed a well written manuscript about an important aspect of clinical cardiology. Interestingly, there is a need for novel biomarkers in order to assess the risk of patients with lipid-disorders. After reading the manuscript I have the following comment to suggest for you:
In the introduction, you could include (in short) the guidelines' recommendations about diagnosis and management of dyslipidemia in these patients.
Response 1: Dear reviewer, we feel very grateful for your recognition of this work. Your comments are critical and useful for improving the quality of our article. According to your suggestions, the guidelines' recommendations about diagnosis and management of homozygous familial hypercholesterolaemia (in short) have been depicted in the section of Introduction as follows: Diagnosis of HoFH can be made on the basis of clinical or genetic criteria. While genetic testing can provide a definitive diagnosis of HoFH. Patients with HoFH usually receive statins in combination with additional cholesterol-lowering agents, such as ezetimibe or PCSK9 inhibitors. Lipoprotein apheresis or liver transplantation may be used as alternative options for patients who have poor response to lipid lowering drugs. Again, thanks a lot for your time and attention in our manuscript. All the related changes were marked up using the “Track Changes” in the revise manuscript and we hope that the revised article can achieve your expectation and will be worthy of publication in Journal of Clinical Medicine.

Reviewer 3 Report
Interesting and timely topic. Paper clearly written.
I only suggest to add a paragraph about the practical applications of these findings.
please try to explain the pathobiology on the basis of the metabolomic difference between the groups
can the results be extended to non-Chinese population?
Author Response
Response to Reviewer 3 Comments
Point 1:
Interesting and timely topic. Paper clearly written.
I only suggest to add a paragraph about the practical applications of these findings.
Please try to explain the pathobiology on the basis of the metabolomic difference between the groups.
Can the results be extended to non-Chinese population?
Response 1: Dear reviewer, thanks a lot for your critical and thoughtful comments. We feel very grateful for your recognition of this work. Your comments are very useful to improve our manuscript. As per your valuable suggestion, we have added a paragraph about the preactical applications of our findings in the section of Discussion as follows: In this study, using rapid and high-throughput LC/MS metabolomic technique, we identified a panel of 21 metabolomic alterations that were independently associated with HoFH condition. We found that HoFH could be correctly preselected in patients with aggressive phenotypes of LDL-C levels by using seven identified metabolite signatures-based diagnostic algorithm, even without genetic tests. Currently, LC/MS-based metabolomic platforms are available in most Grade III Level A hospitals across mainland China. Furthermore, compared with genetic testing, clinical metabolomics is a more economical measure that cost approximately 100–120 dollars. This work demonstrates that the established metabolomic approach may not only provide promising opportunities to accurately differentiate HoFH from the severe form of HeFH in clinical practice, but also offers a more cost-effective option for patients and clinicians.
According to your comments “Please try to explain the pathobiology on the basis of the metabolomic difference between the groups”. We have revised the section of discussion to address your concerns and hope that it is now clearer. In this study, the sera levels of several gut microbiota-derived metabolites (including 3-indolepropionic acid, secondary bile acids), triacylglycerol (TAG) species, and lysophosphatidic acid (LysoPA) species were found to be significantly different between HoFH and severe form of HeFH patients. These altered metabolites have been reported to be closely associated with the cholesterol metabolism and low-density lipoprotein metabolism. At the metabolomic level, our findings may partly explain why the study HeFH patients showed similar LDL-C level phenotype with HoFH patients. However, further studies investigating the mechanism of action of these metabolomic alterations in regulating circulating cholesterol homeostasis are warranted. Another interesting finding of this study is that HoFH patients exhibited higher levels of Lp(a) and its metabolic products (LysoPA species) than patients with severe HeFH. Previous studies have demonstrated the negative functions of Lp(a) and LysoPA in promoting ASCVD progression and regulating inflammation. Therefore, patients with HoFH might have increased ASCVD risk than HeFH patients who also had an aggressive phenotype of LDL-C levels.
According to your comments “Can the results be extended to non-Chinese population?”. This is an very thoughtful comment for us. The ethnic homogeneity of our sample population might limit the generalizability of our findings to other populations. We do hope that include additional validation in other ethnic populations to confirm our results through international cooperation in the future. In a word, we thank very much for your valuable comments, the ethnic homogeneity of our study population has been listed as an important limitation in the section of Discussion in the revised manuscript. Again, thank you very much for your time and attention on our manuscript. All the related changes were marked up using the “Track Changes” in the revise manuscript and we hope that the revised article can achieve your expectation and will be worthy of publication in Journal of Clinical Medicine.
